# Synoptic Analysis of Flood-Causing Rainfall and Flood Characteristics in the Source Area of the Yellow River

**Lijun Jin [1], Changsheng Yan [1], Baojun Yuan [2,\*], Jing Liu [1] and Jifeng Liu [1]**

[1] Hydrology Bureau, Yellow River Conservancy Commission, Zhengzhou 450004, China; jinlj_2020@163.com (L.J.); 13776501048@163.com (C.Y.); liujing0208@163.com (J.L.); jifengliu@163.com (J.L.)

[2] Henan Meteorological Observation Data Center, Henan Meteorological Service, Zhengzhou 450003, China

\* Correspondence: 18538071561@163.com; Tel.: +86-185-3807-1561

**Abstract:** The source area of the Yellow River (SAYR) in China is an important water yield and water-conservation area in the Yellow River. Understanding the variability in rainfall and flood over the SAYR region and the related mechanism of flood-causing rainfall is of great importance for the utilization of flood water resources through the optimal operation of cascade reservoirs over the upper Yellow River such as Longyangxia and Liujiaxia, and even for the prevention of flood and drought disasters for the entire Yellow River. Based on the flow data of Tangnaihai hydrological station, the rainfall data of the SAYR region and NCEP-NCAR reanalysis data from 1961 to 2020, three meteorological conceptual models of flood-causing rainfall—namely westerly trough type, low vortex shear type, and subtropical high southwest flow type—are established by using the weather-type method. The mechanism of flood-causing rainfall and the corresponding flood characteristics of each weather type were investigated. The results show that during the process of flood-causing rainfall, in the westerly trough type, the mid- and high-latitude circulation is flat and fluctuating. In the low vortex shear type, the high pressures over the Ural Mountains and the Okhotsk Sea are stronger compared to other types in the same period, and a low vortex shear line is formed in the west of the SAYR region at the low level. The rain is formed during the eastward movement of the shear line. In the subtropical high southwest flow type, the low trough of Lake Balkhash and the subtropical high are stronger compared to other types in the same period. Flood-causing rainfall generally occurs in areas with low-level convergence, high-level negative vorticity, low-level positive vorticity, convergence of water vapor flux, a certain amount of atmospheric precipitable water, and low-level cold advection. In terms of flood peak increment and the maximum accumulated flood volume, the westerly trough type has a long duration and small flood volume, and the low vortex shear type and the subtropical high southwest flow type have a short duration and large flood volume.

**Keywords:** weather type; circulation; flood-causing rainfall; Yellow River

## 1. Introduction

According to the Sixth Assessment Report of the Intergovernmental Panel on Climate Change (IPCC), climate change and human activities have accelerated the global and regional water cycle, resulting in a continuous increase in the frequency and intensity of extreme climate events [1–4]. Nature disasters, such as floods caused by climate change, pose a serious threat to regional economy and to the safety of people's lives and property [5–10]. In the past 20 years, the global population affected by floods has increased by 20–24%, and the loss caused by floods has reached USD $6510 \times 10^8$ [11,12]. It is predicted that the uncertainty and risk of extreme weather and hydrological events will continue to increase in the future and the resulting flood risk will bring serious obstacles to human sustainable development [13,14]. Therefore, it is urgent to carry out flood-related research, which has important practical significance and long-term value for regional flood risk management, improving water resource utilization efficiency, seeking benefits, and avoiding hazards.

The source area of the Yellow River (SAYR) is located in the northeast of the Tibetan Plateau, which is a flood-prone section of the Yellow River [15,16]. Research data has confirmed that the precipitation and runoff in the SAYR have decreased over the past 60 years but have increased since 2000 [17–21]. Simulation results indicate that flooding in the region may enter a medium-to-large state in the future [22–24]. Therefore, from the perspective of disaster prevention and reduction, strengthening the study of rainfall and flood in the SAYR, and accurately forecasting and taking preventive measures in advance are not only related to reducing property losses and casualties, but also an urgent need for flood resource management in cascade reservoirs such as Longyangxia and Liujiaxia.

As the first driving element of flood in the SAYR, rainfall has been a matter of deep concern by meteorologists. Studies have shown that there are two typical types of rainfall in the SAYR: continuous rainfall and heavy rainfall, which are mainly influenced by the westerly circulation system, the East Asian monsoon system, and the plateau system [25,26]. The South Asian high, subtropical westerly jet, short-wave troughs, and shear lines are the main weather systems that affect the short-term heavy rainfall in the region. In addition, short-term heavy rainfall of the low trough type is common in late spring, early summer, and autumn, while short-term heavy rainfall of the southwest airflow type is common in summer [27–29]. The abnormality of the South Asian high and Western Pacific subtropical high are the most prominent features of atmospheric circulation in the continuous rainfall [30]. When the Western Pacific subtropical high is stronger and westward, it is conducive to the strengthening of water vapor transport along the southern boundary, resulting in increased rainfall and more abundant runoff in the SAYR [31]. Moreover, when the westerly jet of the Asian subtropical zone moves southward, it is conducive to the formation of abnormal cyclones over Central Asia, which also easily leads to more rainfall in the upper reaches of the Yellow River [32].

Rainfall in the Yellow River is not only affected by the local climate systems, but also by external forcing factors. Therefore, some studies have analyzed the influence of remote correlation of global atmospheric circulation and SST field on rainfall. The results show that sunspots, polar vortices, NAO, ENSO, and Indian Ocean SST cause rainfall anomalies in the SAYR by interacting with the atmospheric circulation [33–39]. Studies of paleo-flood events show that the abrupt or turning stage of climate is prone to flood. At this time, the atmospheric circulation of the East Asian monsoon is extremely unstable, with an increase in the rate of climate variability and the extremity of rainfall [40].

To sum up, current studies on rainfall in the SAYR either analyze its weather and climate causes from the perspective of rainfall without considering floods or analyze its rainfall characteristics and corresponding meteorological causes based on the large flood process over the past years, and rarely analyze the synoptic mechanism of flood-causing rainfall by linking rainfall with floods at all levels, especially small and medium floods. Therefore, in order to improve the research on flood-causing rainfall in the SAYR, this paper investigates and analyzes flood cases at all levels in the SAYR in the past 60 years, classifies the flood-causing rainfall process, reveals the similarities and differences of circulation characteristics and physical quantities corresponding to different types, and discusses the flood characteristics formed under different weather types, so as to provide more scientific basis for disaster prevention and reduction in the Yellow River.

## 2. Data and Methods

### 2.1. Study Area

The SAYR region refers to the area of the main stream of the Yellow River above the Tangnaihai hydrological station, ranging from 95°30′–103°30′ E, 32°10′–36°05′ N. The SAYR drainage area is approximately 122,000 km$^2$, accounting for 16% of the total Yellow River drainage area, with an average annual natural runoff of 20.5 billion m$^3$. It accounts for about 38% of the total natural runoff of the Yellow River and is known as the "Yellow River water tower" [41]. The SAYR has a plateau continental climate, strongly influenced by the southwest monsoon, with annual precipitation ranging from 250 mm to 750 mm [42,43].

### 2.2. Data

This study used daily observational rainfall from 12 national meteorological stations (black triangles in Figure 1) in the SAYR region, which was obtained from the National Meteorological Information Center (NMIC) of the China Meteorological Administration and was available at http://data.cma.gov.cn/ (1 March 2023). This data has undergone strict quality control by NMIC, including internal consistency checks, spatial and temporal consistency checks, and the identification of extreme values [17]. Besides, the missing data was extended based on the interpolation method in the study.

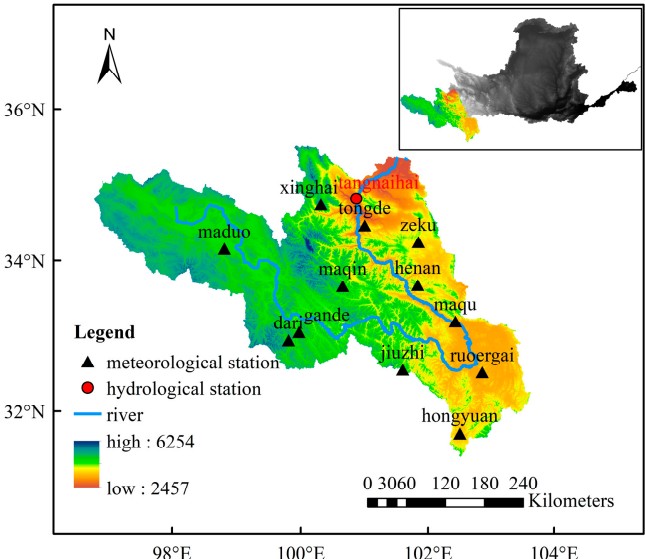

**Figure 1.** The geographical location and the distribution map of 12 meteorological stations and Tangnaihai hydrological station in the SAYR.

Tangnaihai is an important hydrological station in the Yellow River, which has been considered as the outlet of the SAYR. It plays a very important role in flood control dispatching and unified management of water resources on the Yellow River. Therefore, the flow at Tangnaihai can represent the flow of the whole SAYR. The daily average flow data of Tangnaihai was obtained from the Hydrology Bureau of Yellow River Conservancy Commission. It is an official and authoritative hydrological unit, and the data is of high enough quality after being reviewed by many parties.

The study also used daily mean pressure-level geopotential height, U- and V-wind, specific humidity, pressure, temperature and atmospheric precipitable water by the National Centers for Environmental Prediction (NCEP)-National Center for Atmospheric Research (NCAR) reanalysis data set [44]. The NCEP–NCAR reanalysis has a horizontal resolution of $2.5 \times 2.5°$.

The above data are all extracted from 1961 to 2020. The flow and rainfall data are used to define the flood process and the flood-causing rainfall process, and to investigate the corresponding flood characteristics of each type over the SAYR. The NCEP–NCAR data is used to analyze the synoptic mechanism of the flood-causing rainfall process.

### 2.3. Definition of Flood

According to the "Regulations on Flood Numbering of Major Rivers in China" issued by the Ministry of Water Resources [45], flood numbering is carried out when the flow of the Tangnaihai hydrological station (also known as the Longyangxia reservoir intake station) reaches 2500 m³/s. Referring to the operation scheme of the reservoir in the upper reaches of the Yellow River of the latest 10a, when the water level of Longyangxia reservoir is below the designed flood limit of 2594 m, the flow should be controlled by no more than 2000 m³/s, which can take into account the flood-control requirements of the lower reaches

of the river. At the same time, considering the actual operation, when the water level of Longyangxia reservoir is below the flood limit of 2588 m, the water will not be abandoned according to the power generation, and the reservoir will be discharged at the flow rate of 1000–1500 m³/s.

In summary, when the flood peak at Tangnaihai station reaches 2500 m³/s or above, it is marked as a large flood. When the flood peak reaches 2000–2500 m³/s, it is marked as a medium flood, and when the flood peak reaches 1500–2000 m³/s, it is marked as a small flood. According to the above flood standards, a total of 85 floods occurred in the SAYR from 1961 to 2020.

### 2.4. Definition of Rainfall Process

According to the forecasting experience of the flood in the upper reaches of the Yellow River by the Hydrology Bureau of Yellow River Conservancy Commission, it is generally believed that the flood peak will appear at Tangnaihai station 3–7 days after the maximum 5 day moving average rainfall appears in the SAYR. In view of this, the specific method of defining the flood-causing rainfall process is as follows. Firstly, the daily rainfall of the SAYR is averaged from 12 meteorological stations by the Tyson polygon method [46]. The 5 day moving average rainfall of the SAYR refers to an average rainfall of the day and the previous four days. Effective rainfall is defined as daily rainfall with 1 mm or more of the SAYR, and otherwise it is ineffective rainfall. Secondly, find the time when the flood peak appears (T1), and according to T1, find the time when the maximum 5 day moving average rainfall appears within 3–7 days (T2). On the basis of the 5 day moving average rainfall definition, the start time of the flood-causing rainfall process is tentatively determined by pushing forward four days of T2, and T1 is taken as the end time of the rainfall process. Thirdly, if there is effective rainfall on the day before the start time, the rainfall process will extend forward by one day and continue to search forward until there is no effective rainfall on the previous day, and it will ultimately be set as the start time of the rainfall process. Finally, starting from T2 and searching backwards until the next day when there is no effective rainfall, it is ultimately determined as the end time of the rainfall process, during which there can be at most one day of ineffective rainfall. The schematic diagram of the method of defining the flood-causing rainfall process is shown in Figure 2.

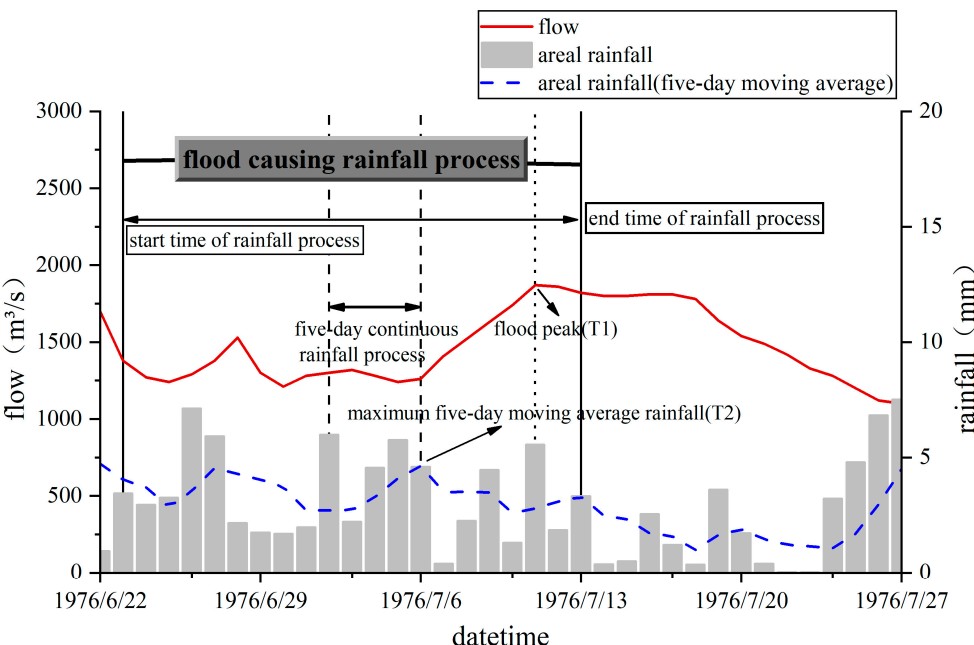

**Figure 2.** The schematic diagram of the method of defining the flood-causing rainfall process.

*2.5. Methods*

According to 500 hPa circulation, the flood-causing rainfall was classified to three weather types: westerly trough type (53%), low vortex shear type (14%), and subtropical high southwest flow type (33%). To avoid flattening circulation anomalies after averaging multiple cases, 3 typical processes (27 in total) were selected for each weather type according to the different flood magnitudes. The 200 hPa height field and wind field, 500 hPa height field and anomaly field were synthesized and analyzed. In addition, the violin plot method was used to analyze the dynamics, water vapor and instability conditions of each rainfall process. The average values of vorticity, divergence, atmospheric precipitable water, vapor flux divergence, and temperature advection values of the SAYR (referring to 32.5–37.5° N, 95–105° E) of each process were displayed by the violin plot, and the physical quantities of different floods were tested to see whether there were significant differences. The violin plot is a combination of box plot and kernel density plot, which can display the overall distribution characteristics and probability density of multiple sets of data. The profile width of the piano body represents the probability density of distribution, and the wider the profile is, the more concentrated the data is at this location, resulting in a higher probability density. Meanwhile, the distribution of the upper quartile (25% box), median, lower quartile (75% box), mean of the data can be integrated and displayed.

## 3. Results

*3.1. Circulation Background of Three Weather Types*

3.1.1. Westerly Trough Type (Type I)

When type I flood-causing rainfall occurs, there are obvious differences in the circulation corresponding to different flood magnitudes. In general, as the flood magnitude increases, the 200 hPa South Asian high (SAH) gradually strengthens, the ridge line moves northward, and the area of the SAH expands eastward (Figure 3a,c,e). The maximum central intensity of the SAH corresponding to the small flood is 1248 dagpm, and the ridge line is located at 23° N. The central intensity of the SAH corresponding to the medium and large floods exceeds 1252 dagpm, and the ridge lines are located at 28° N and 30° N, respectively, which are slightly northward and eastward compared to the average location of the SAH. The figure also shows that the maximum wind speed of the jet core corresponding to different floods exceeds 35 m/s. The SAYR is located on the right side of the jet entrance area, with strong high-altitude divergence, which is conducive to large-scale upward movement.

The latitude circulation prevails at 500 hPa, with multiple short-wave trough fluctuations. The troughs are located near the Ural Mountains and the Sea of Japan (Figure 3b,d,f). The Western Pacific subtropical high (WPSH) lies to the east and south, and the 588 dagpm line corresponding to the small flood cannot be analyzed in the height field, but the Shandong Peninsula shows a positive height abnormality, which corresponds to the strong continental high. The main part of the WPSH, corresponding to the medium and large floods, is located at sea, and the 584 dagpm line around the WPSH is located from the middle and lower reaches of the Yangtze River to South China. When rainfall occurs, the short-wave trough that splits from the Ural Mountains trough moves eastward and southward along the northern Xinjiang, guiding cold air to penetrate southward. At the same time, the Indo-Myanmar trough located in the Indian Peninsula is stronger compared to other types in the the same period of the year. The SAYR is controlled by the southwesterly air in front of the Indo-Myanmar trough and the periphery of the WPSH, which intersects with the cold air in the north and forms continuous rainfall.

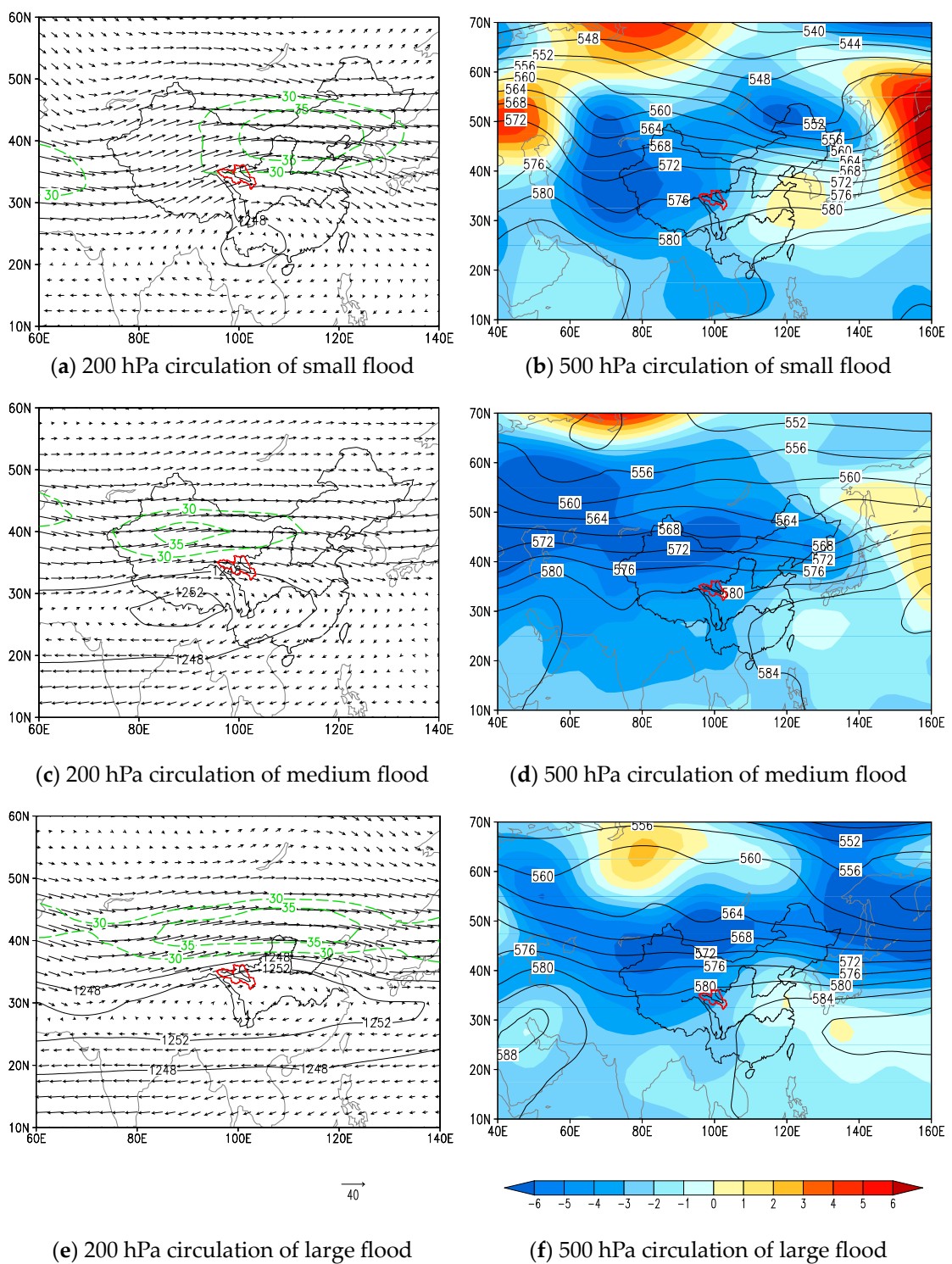

**Figure 3.** (**a**) Geopotential height field (black line, unit: dagpm), wind field (vector field, unit: m/s), and upper jet stream (green line, unit: m/s) at 200 hPa of a small flood when the type I flood-causing rainfall occurs; (**c**,**e**) as in (**a**), but for a medium flood and a large flood. (**b**) Geopotential height field (black line, unit: dagpm) and its anomaly (color-filled, unit: dagpm) at 500 hPa of a small flood; (**d**,**f**) as in (**b**), but for a medium flood and a large flood.

### 3.1.2. Low Vortex Shear Type (Type II)

When type II flood-causing rainfall occurs, both the intensity and the area of the SAH are significantly stronger than those of type I (Figure 4a,c,e). The maximum central intensity

of the SAH exceeds 1252 dagpm, especially the large flood, which exceeds 1260 dagpm. The axis of the upper-level jet is located near 40° N, which is not much different from type I. The main difference is that the distribution of the medium and large flood jet is northeast-southwest, while that of type I is mainly east-west. In summary, the main characteristic of type II is that the SAH is stronger than type I, and the trend of the jet has changed. The SAYR is located to the right side of the jet's entrance area, with strong high-altitude divergence, which is conducive to large-scale upward motion.

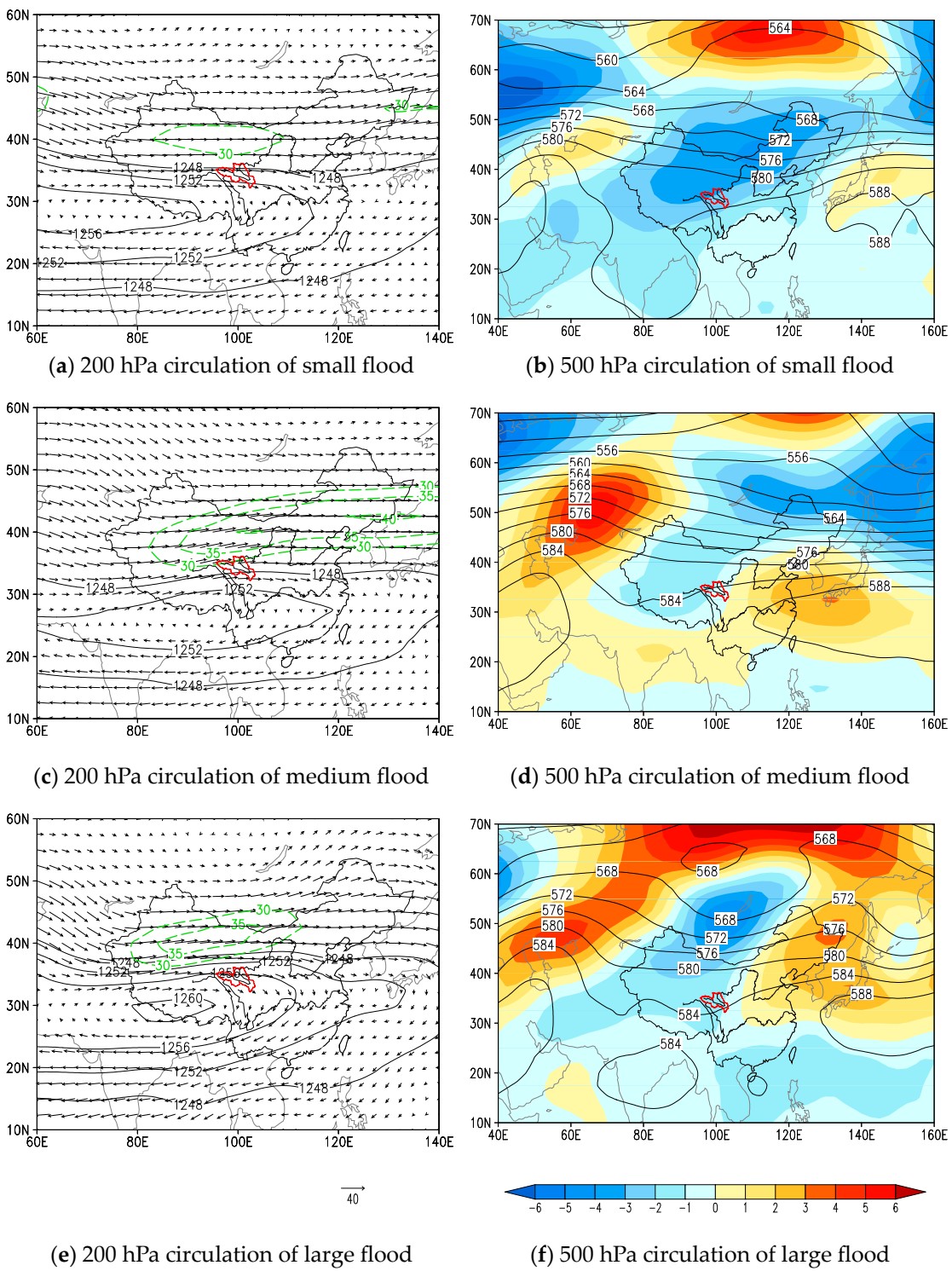

(**a**) 200 hPa circulation of small flood      (**b**) 500 hPa circulation of small flood

(**c**) 200 hPa circulation of medium flood      (**d**) 500 hPa circulation of medium flood

(**e**) 200 hPa circulation of large flood      (**f**) 500 hPa circulation of large flood

**Figure 4.** As in Figure 3, but for when the type II flood-causing rainfall occurs.

At 500 hPa, a meridional circulation of two ridges and one trough is present, with the Ural Mountains and the Sea of Okhotsk as a high-pressure ridge and Lake Baikal as a low-pressure trough (Figure 4b,d,f). The low vortex shear line formed between the two highs lies to the west of the SAYR. The main body of the WPSH is still located at sea, but it extends more westward than type I. For example, in the medium flood, the west ridge point of the WPSH is located near 110° E, the ridge line of the WPSH is located north of 28° N, and the 584 dagpm line is lifted to the Sanhua (which is located in the middle reaches of the Yellow River)-Weihe-the SAYR. When rainfall occurs, the low trough originally located in Xinjiang moves eastward into the Inner Mongolian-Hetao area. The northerly wind in front of the Ural ridge guides the cold air southward, which together with the southwesterly flow in front of the upper trough and the WPSH intensifies the development of the low-level shear line, resulting in rainfall on the eastern side of the shear line. Compared to type I, the meridional extent of the mid- and high-latitude circulation in type II increases significantly, and the WPSH extends westward and northward and strengthens.

### 3.1.3. Subtropical High Southwest Flow Type (Type III)

When type III flood-causing rainfall occurs, the intensity and morphological characteristics of the SAH corresponding to different floods are similar. As can be seen from Figure 5a, the maximum intensity of the SAH in the small flood exceeds 1252 dagpm, which exceeds 1256 dagpm in the medium and large flood, and the ridge line of the SAH is located at 28–30° N (Figure 5c,e). From the distribution of the upper-level jet, the maximum wind speed exceeds 35 m/s in the small flood and medium flood, and that exceeds 45 m/s in the large flood, which is the strongest upper-level jet in all flood-causing weather types, and the jet axis is located at 40° N. Therefore, the upper-level circulation corresponding to the type III flood-causing rainfall is consistent, characterized by the overall strong and eastward SAH, which is similar to type II and stronger than type I. In particular, the upper-level jet in the large flood is unusually strong, and the SAYR is located to the right side of the upper-level jet entrance area, which is conducive to the generation of divergence and upward motion and the formation of rainfall.

The 500 hPa circulation consists of two troughs and two ridges, of which two ridges are located in the Ural Mountains and Lake Baikal, respectively, and two troughs are located in Lake Balkhash and Northeast China (Figure 5b,d,f). Type III is obviously different from type I and type II in the WPSH. The WPSH of type III extends to near 115° E, the ridge line is located at 27–28° N, the 588 dagpm line is stable on the coast of East China, and the 584 dagpm line is located in the Sanhua-Weihe-the SAYR. The Balkhash Lake trough and the Indo-Myanmar trough are stronger compared to the other types in the same period (the height is negative anomaly), and the southwest flow in front of the trough and the flow on the west side of the WPSH are superimposed in the SAYR, causing rainfall. Therefore, in type III, there is not much difference in the circulation of different floods. The main difference from the other two types of circulation is that the WPSH extends westward and is stronger, providing abundant water vapor and dynamic conditions for rainfall in the SAYR.

### 3.2. Characteristics of Physical Quantity of Three Weather Types

The mean divergence (200 hPa, 500 hPa, 700 hPa), vorticity (200 hPa, 500 hPa, 700 hPa), total water vapor flux divergence, atmospheric precipitable water, and temperature advection (500 hPa, 700 hPa) of the SAYR during 39 rainfall days of small flood, 22 rainfall days of medium flood, and 33 rainfall days of large flood of type I, 34 rainfall days of small flood, 32 rainfall days of medium flood, and 29 rainfall days of large flood of type II, and 32 rainfall days of small flood, 38 rainfall days of medium flood, and 41 rainfall days of large flood of type III are counted, and the violin plot is obtained by arranging the groups.

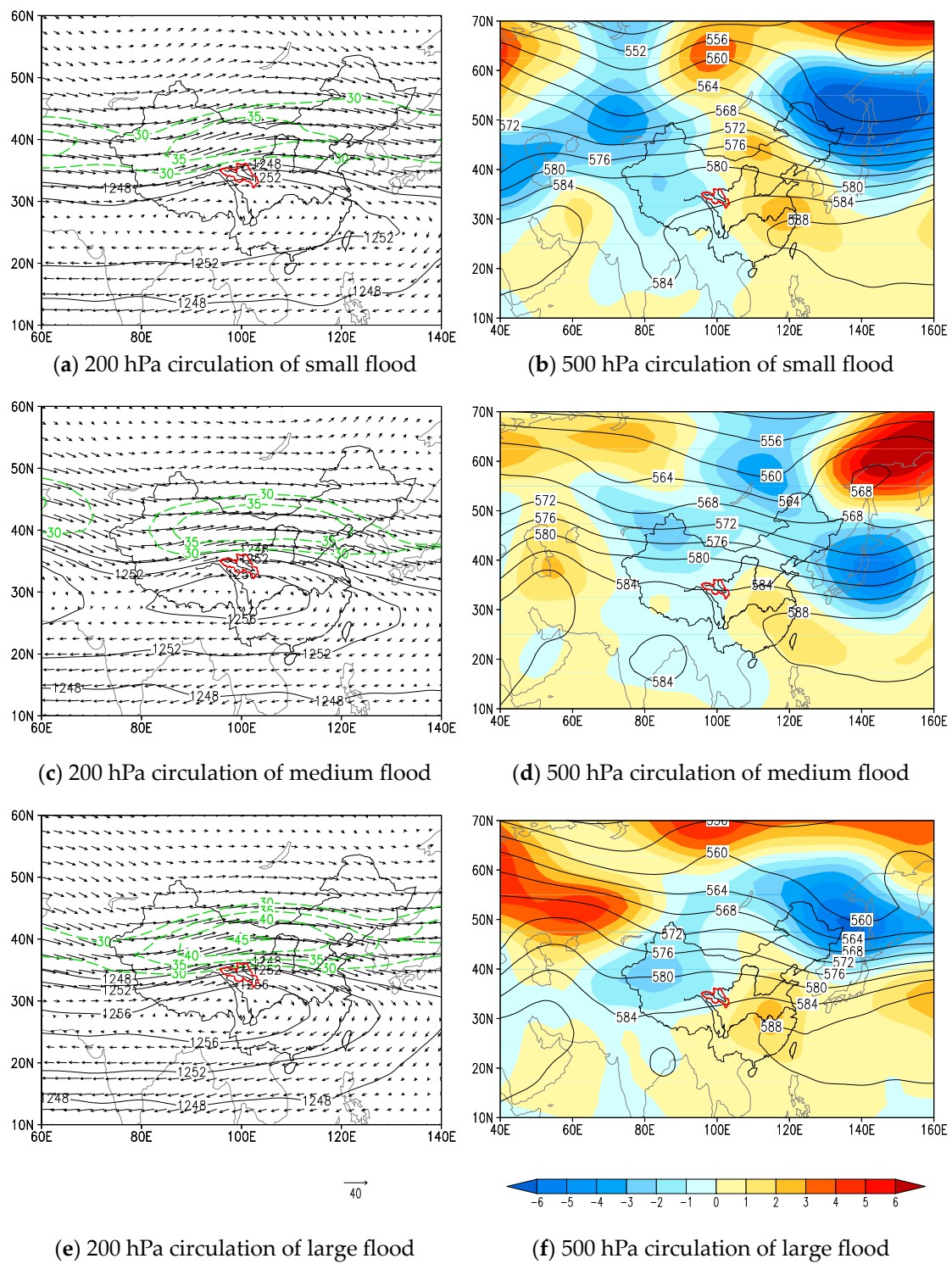

**Figure 5.** As in Figure 3, but for when the type III flood-causing rainfall occurs.

As can be seen from Figure 6, the mean value of the 200 hPa divergence is positive for type I flood-causing rainfall, but the dispersion is relatively large. There are significant differences between the large flood and small (medium) flood. The 500 hPa divergence is similar, but there is no significant difference between different floods. The mean, extreme, and quartile values of 700 hPa are all below 0. The convergence of the medium flood is the strongest, with a mean value of $-5.47 \times 10^{-6}$ s$^{-1}$. There are significant differences between the large flood and small (medium) flood. In the vorticity field, both the mean

and the quartile values of 200 hPa are negative, with significant differences between large flood and small (medium) flood. The mean value of the 500 hPa vorticity is negative and the dispersion is also large. Significant differences exist between the medium flood and large flood. The mean and quartile values of 700 hPa correspond to positive vorticity and there are significant differences between the small flood and medium flood. In the water vapor field, the water vapor condition of the small flood is general; in particular the mean value of total water vapor flux divergence is positive, which is manifested as water vapor divergence. The water vapor condition of the large flood is optimal, with the mean value of water vapor flux divergence of $-3.42 \times 10^{-8}$ g·cm$^{-2}$·hPa$^{-1}$·s$^{-1}$ and the quartile range of $(-7.77 \sim -0.08) \times 10^{-8}$ g·cm$^{-2}$·hPa$^{-1}$·s$^{-1}$. The atmospheric precipitable water condition of the large flood is more adequate than the other two types of floods, but there are no significant differences between different floods. In the thermal field, the temperature advection of 500 hPa and 700 hPa is relatively dispersed, the indicative significance is not strong, and especially the temperature advection of small flood is weak. It can be seen that the flood-causing rainfall occurs in the region with convergence of low level, divergence of high level, convergence of water vapor flux and a certain atmospheric precipitable water. The difference in vapor flux divergence between different floods is the most significant, followed by 200 hPa divergence and 200 hPa vorticity.

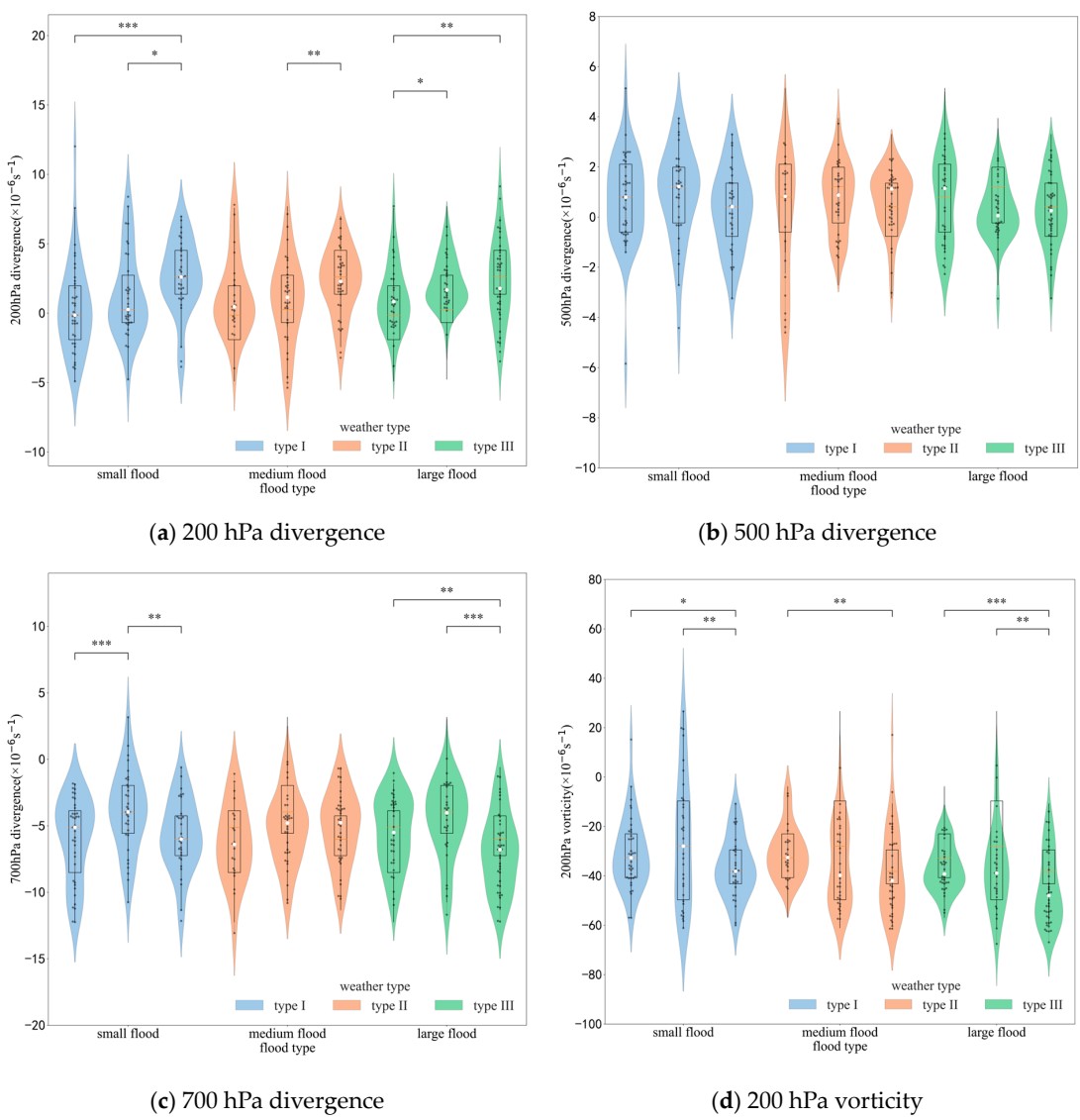

(**a**) 200 hPa divergence

(**b**) 500 hPa divergence

(**c**) 700 hPa divergence

(**d**) 200 hPa vorticity

**Figure 6.** *Cont.*

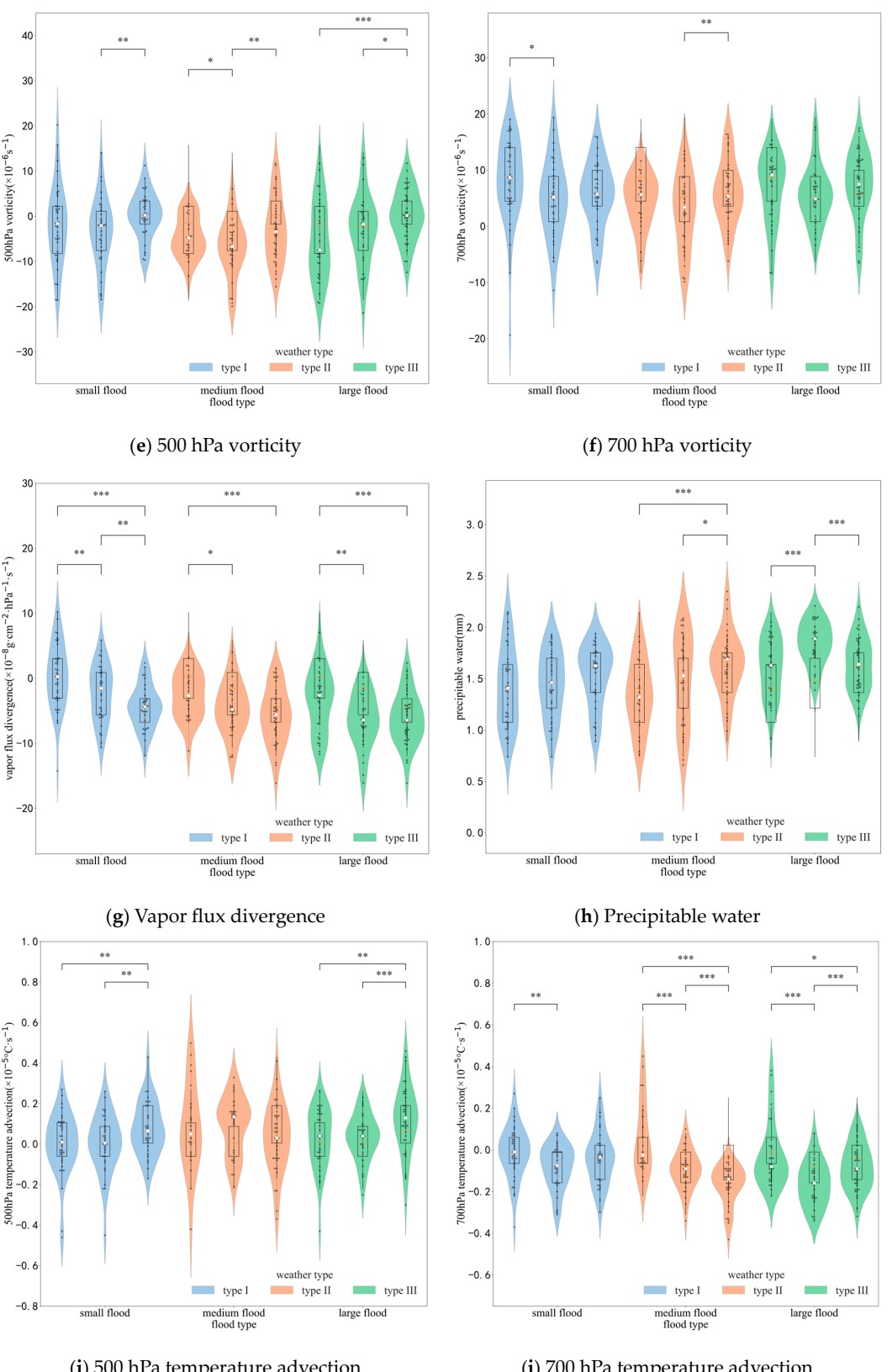

(**e**) 500 hPa vorticity

(**f**) 700 hPa vorticity

(**g**) Vapor flux divergence

(**h**) Precipitable water

(**i**) 500 hPa temperature advection

(**j**) 700 hPa temperature advection

**Figure 6.** Violin plot of the physical quantity characteristic statistics of various flood types in the SAYR region. (The bottom and top of the vertical black bars in the violin plot indicate the 25th and 75th percentiles, the white circles indicate the mean values, and the different symbols marked on the top of the violin plot indicate that the mean values of physical quantities are significantly different, ***, $p < 0.001$, **, $p < 0.05$, *, $p < 0.1$).

When type II flood-causing rainfall occurs, 200 hPa is a divergence field, which is obviously stronger than that of type I, especially for the large flood, with a mean value of $2.12 \times 10^{-6}$ s$^{-1}$ and a quartile range of $(0.79 \sim 3.25) \times 10^{-6}$ s$^{-1}$. There are significant differences between the medium flood and large flood. A pressure of 500 hPa is mainly divergent, but the dispersion is large. A pressure of 700 hPa shows convergence, and the intensity is weaker in both mean and quartile values compared to type I, of which there is no significant difference between different floods. In the vorticity field, 200 hPa is negative, and the dispersion of medium and large floods is larger than that of type I. There are significant differences between the small flood and large flood. The 500 hPa vorticity is similar to type I, and the dispersion is larger, with significant differences between the medium flood and small (large) flood. A pressure of 700 hPa is a positive vorticity field, and the mean value is weaker than that of type I, of which there are significant differences between the medium flood and large flood. The water vapor conditions of different floods are relatively favorable. As the flood magnitude increases, the water vapor conditions tend to be better, and there are significant differences between the small flood and medium (large) flood. The divergence of the water vapor flux in the medium and large flood is better than that in type I, both in the mean and quartile values. The same is true for atmospheric precipitable water. In the thermal field, the 500 hPa temperature advection is weak and the dispersion is large, and the 700 hPa temperature advection is negative in both the mean and quartile values, which is better than that of type I, showing cold advection, and there are significant differences between different floods. This shows that the dynamic conditions of the lower level of type II flooding rainfall are weaker than those of type I, but the dynamic conditions of the upper level are stronger than those of type I, and the conditions of water vapor and 700 hPa cold advection are also stronger than those of type I. The difference of 700 hPa advection between different floods is the most significant, followed by the divergence of vapor flux and precipitable water.

For type III flood-causing rainfall, the 200 hPa divergence is better than that of type I and type II, which is a good indicator for forecasting, and both the mean and the quartile values are positive. In addition, there are significant differences between the small flood and medium (large) flood. The dispersion of 500 hPa is still large, with no significant difference between different floods. The 700 hPa convergence is also better than that of type I and type II, especially for the large flood, with a mean value of $-7.01 \times 10^{-6}$ s$^{-1}$ and a quartile range of $(-9.52 \sim -5.20) \times 10^{-6}$ s$^{-1}$, which is the strongest among all flood-causing weather types. There are significant differences between the large flood and small (medium) flood. In the vorticity field, the 200 hPa negative vorticity and 700 hPa positive vorticity are also better than the other two types, and the dispersion of 500 hPa is large, which is also the case in the other two types. The total vapor flux divergence and precipitable water of the small and medium flood of type III are better than those of type I and type II, but the vapor condition of the large flood is worse than that of type II. There are significant differences between the small flood and medium (large) flood in vapor flux divergence, with significant differences between the medium flood and small (large) flood. In the thermal field, the 500 hPa temperature advection is not significantly different from type I and type II, showing a better indication in single levels and single magnitude floods. The 700 hPa shows cold advection of the large flood, with a mean value of $-0.08 \times 10^{-5}$ °C·s$^{-1}$, which is weaker than a type II flood of the same magnitude. In addition, there are significant differences between different floods. This shows that the dynamic conditions of type III flood-causing rainfall are the strongest of the three types, the water vapor conditions and thermal conditions of the small and medium floods are the strongest of the three types, and the water vapor conditions and thermal conditions of the large flood are between type I and type II. The difference in 700 hPa advection between different floods is the most significant, followed by the difference in vapor flux divergence and precipitable water.

Through the analysis of the physical quantity fields at high and low altitudes, the physical quantity values that meet the 75% ratio (i.e., the upper and lower quartile values)

and have the same symbol as the mean value are counted as the threshold values, and the quantitative indicators of the diagnosis and analysis of the physical quantity of flood-causing rainfall with beneficial indicative significance are determined, as shown in Table 1. It can be seen that in the dynamic field, the 700 hPa divergence and vorticity, and the 200 hPa vorticity generally have good indicative significance, and the 200 hPa divergence of type III has good indicative significance. In the water vapor field, the indication of atmospheric precipitable water is good, and the total water vapor flux divergence is also favorable for type II and type III floods. In the thermal field, the 700 hPa temperature advection is a good indicator of type II and type III floods. Other physical quantities only show excellent indicative significance only for single levels and single magnitude floods. This is consistent with the general theory of rainfall weather that can be used for daily forecasting.

**Table 1.** Threshold of physical quantity of flood-causing rainfall in the SAYR region.

| | | $D_{iv200}$ ($\times 10^{-6}$ s$^{-1}$) | $D_{iv500}$ ($\times 10^{-6}$ s$^{-1}$) | $D_{iv700}$ ($\times 10^{-6}$ s$^{-1}$) | $V_{or200}$ ($\times 10^{-6}$ s$^{-1}$) | $V_{or500}$ ($\times 10^{-6}$ s$^{-1}$) | $V_{or700}$ ($\times 10^{-6}$ s$^{-1}$) | VF ($\times 10^{-8}$ g·cm$^{-2}$·hPa$^{-1}$·s$^{-1}$) | PW (mm) | $T_{500}$ ($\times 10^{-5}$ °C·s$^{-1}$) | $T_{700}$ ($\times 10^{-5}$ °C·s$^{-1}$) |
|---|---|---|---|---|---|---|---|---|---|---|---|
| Type I | Small flood | - | - | −3.66 | −22.18 | - | 4.31 | - | 1.07 | - | - |
| | Medium flood | - | - | −4.19 | −25.80 | - | 1.39 | - | 1.08 | - | - |
| | Large flood | - | - | −3.23 | −30.59 | - | 2.15 | −0.08 | 1.36 | - | - |
| Type II | Small flood | - | - | −1.83 | −9.03 | - | 0.31 | - | 1.18 | - | −0.01 |
| | Medium flood | - | - | −3.64 | −21.31 | −3.40 | - | −1.86 | 1.05 | - | −0.04 |
| | Large flood | - | - | −1.98 | −27.78 | - | 2.62 | −2.78 | 1.74 | - | −0.10 |
| Type III | Small flood | 1.18 | - | −4.15 | −29.24 | - | 3.28 | −2.79 | 1.36 | - | −0.03 |
| | Medium flood | 1.17 | - | −3.49 | −26.98 | - | 2.57 | −2.90 | 1.46 | - | −0.10 |
| | Large flood | 0.28 | - | −5.20 | −32.63 | - | 4.94 | −4.10 | 1.44 | 0.03 | −0.01 |

### 3.3. Synoptic Model of Flood-Causing Rainfall

Based on the circulation background of the above three flood-causing rainfall weather types, as well as the physical quantity conditions of dynamic, water vapor, and thermal fields, three conceptual models of flood-causing rainfall are established.

The type I synoptic model is shown in Figure 7a. The ridge line of the 200 hPa SAH is located at 23–28° N, and the maximum intensity of the center exceeds 1252 dagpm. The latitudinal circulation is dominant at 500 hPa. The short-wave trough in northern Xinjiang moves eastward and southward, guiding cold air to penetrate southward, and the Indo-Myanmar depression is strong, guiding the low-level warm and wet air to be transported into the SAYR, forming continuous rainfall.

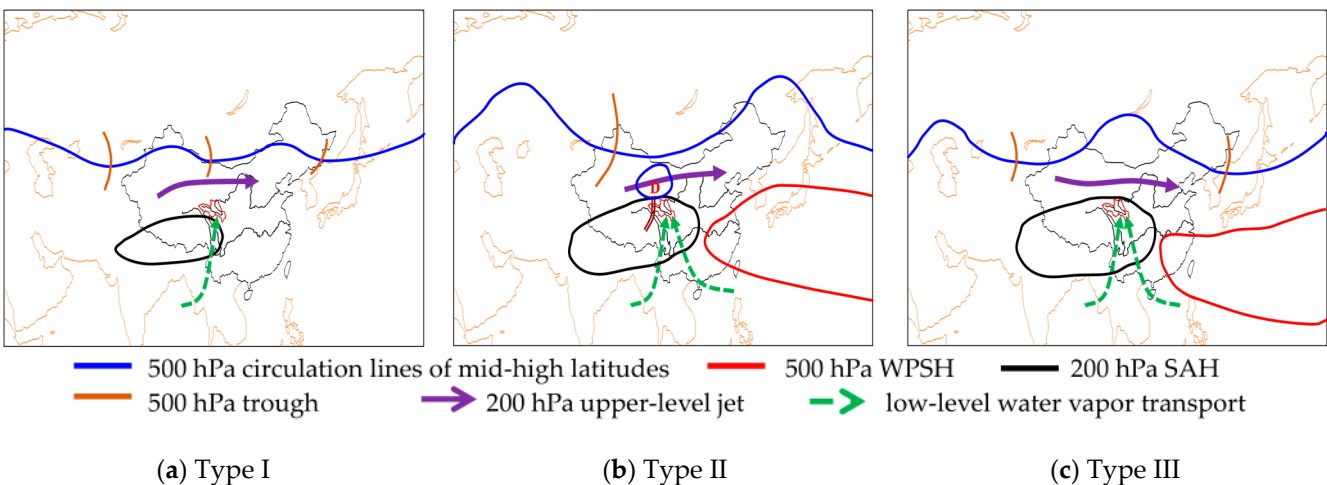

(**a**) Type I          (**b**) Type II          (**c**) Type III

**Figure 7.** Conceptual models of flood-causing rainfall in the SAYR region.

The type II synoptic model is shown in Figure 7b. The ridge line of the 200 hPa SAH is located at 28–30° N, and the maximum intensity of the central is 1252–1256 dagpm. The area of the SAH is obviously larger than that of type I, the location extends eastward, and

the upper-level jet mostly shows a northeast-southwest direction. A pressure of 500 hPa appears as two ridges and a trough with large circulation radiality. The WPSH has a strong force and a northerly ridge, and is mainly located near 28° N. As the Balkhash Lake trough moves eastward, a low vortex is formed over the basin along with a north–south shear line, resulting in rainfall on the east side of the shear line.

The type III synoptic model is shown in Figure 7c. The ridge line of the 200 hPa SAH is located at 28–30° N, with a maximum central intensity over 1256 dagpm. The upper-level jet is the strongest of the three types. A pressure of 500 hPa presents two ridges and two troughs, with large circulation radiality, similar to type II. The WPSH has a strong force, and the ridge line is located at 27–28° N. At the same time, the WPSH is located to the west, providing abundant water vapor and dynamic conditions for rainfall.

### 3.4. Flood Characteristics of Three Weather Types

There are obvious differences in the flood characteristics caused by different weather types. Table 2 shows the flood characteristics corresponding to the three weather types. In general, as the flood magnitude increases in the same weather type, the flood characteristic values also increase. In type I, the average flood peak, process flood volume, maximum three-day flood volume, and maximum three-day flood volume corresponding to the small flood are the smallest among all types, only 1521 $m^3$/s, 11.7 × 10$^8$ $m^3$, 3.6 × 10$^8$ $m^3$, and 5.8 × 10$^8$ $m^3$, and the rainfall duration is the longest, up to 13 days. This type of small flood has the characteristics of long duration and small flood volume, which is related to the long-term continuous rainfall and weak rainfall intensity caused by the disturbance of small trough. The average flood peak, process flood volume, maximum three-day flood volume and maximum five-day flood volume of type I of medium and large floods are moderate among all types. The process flood volume of medium flood is 16.2 × 10$^8$ $m^3$, which is larger than that of type II and type III, indicating that although the rainfall intensity of type I is generally weak, there are many heavy rainfall days during the period, resulting in a larger cumulative flood volume.

**Table 2.** Flood characteristics of three weather types.

|  |  | Flood Peak (m$^3$/s) | Process Flood Increase (m$^3$/s) | Process Flood Volume (×10$^8$ m$^3$) | Maximum 3 Day Flood Volume (×10$^8$ m$^3$) | Maximum 5 Day Flood Volume (×10$^8$ m$^3$) | Duration (days) |
|---|---|---|---|---|---|---|---|
| Type I | Small flood | 1521 | 801 | 11.7 | 3.6 | 5.8 | 13 |
|  | Medium flood | 2102 | 811 | 16.2 | 5.1 | 8.1 | 12 |
|  | Large flood | 2709 | 1500 | 22.2 | 6.8 | 10.7 | 13 |
| Type II | Small flood | 1645 | 492 | 13.0 | 4.1 | 6.5 | 10 |
|  | Medium flood | 2203 | 906 | 14.6 | 5.4 | 8.4 | 10 |
|  | Large flood | 2762 | 1086 | 23.7 | 6.7 | 10.5 | 13 |
| Type III | Small flood | 1643 | 615 | 12.0 | 4.1 | 6.4 | 10 |
|  | Medium flood | 2211 | 1065 | 20.0 | 5.3 | 8.4 | 12 |
|  | Large flood | 2827 | 1505 | 20.2 | 6.8 | 10.5 | 12 |

In type II, the small flood and medium floods show the characteristics of short duration, large flood peak, and large maximum three-day flood volume and five-day flood volume, which are related to the rapid moving speed of the system and high rainfall intensity. The increase in the small flood is only 492 $m^3$/s, which is the smallest of all types. The flood peak and process increase in type II of large floods are 2762 $m^3$/s and 1086 $m^3$/s, respectively, which are smaller than those of type I and type III, and the process flood volume is 23.7 × 10$^8$ $m^3$, which is larger than those of type I and type III. The maximum three-day and five-day flood volumes are close to those of the three weather types. This type of flood is characterized by short duration and large flood volume.

In type III, the average flood peak, process increase, process flood volume, and maximum three-day and five-day flood volumes of the small flood are between type I and type II, which also show the characteristics of short duration. The average flood peak, process increase, process flood volume, and maximum three-day and five-day flood volumes of medium flood of type III are 2211 $m^3/s$, 1065 $m^3/s$, 20.0 × 10$^8$ $m^3$, 5.3 × 10$^8$ $m^3$, and 8.4 × 10$^8$ $m^3$, respectively, which are larger than those of type I and type II, showing the characteristics of great flood peak, flow increase, and flood volume. The average flood peak, process increase, and maximum three-day flood volume of the large flood of type III are the largest among all weather types, reaching 2827 $m^3/s$, 1505 $m^3/s$, and 6.8 × 10$^8$ $m^3$, respectively. In general, the flood peak and cumulative flood volume of this type of large flood are the largest, the medium flood is weak, and the small flood is moderate. The duration of the process is also between type I and type II, which is closely related to the fact that the rainfall at the edge of the WPSH is stronger than the rainfall caused by the moving trough and weaker than the rainfall caused by the confrontation of two highs.

## 4. Discussion

At present, most studies on regional rainfall in China take administrative regions as the unit. However, the SAYR region spans the Qinghai, Sichuan, and Gansu provinces, and there may be large differences in rainfall characteristics and causes in different regions. Therefore, it is necessary to conduct rainfall research on specific regions. Current studies on rainfall in the SAYR mostly focus on short-term heavy rainfall or a specific rainfall process, but there are few studies that combine rainfall and floods to study flood-causing rainfall. In this paper, flood-causing rainfall in the SAYR is classified into three weather types according to the 500 hPa atmospheric circulation, which is basically consistent with the classification of heavy rainfall in Qinghai Plateau summarized by predecessors. On this basis, the circulation differences of floods of different magnitude under each weather type are revealed. Overall, the SAH corresponding to the large flood is stronger and extends eastward than that of the medium and small flood. At the same time, the upper-level jet is also stronger and extends northward, resulting in better conditions and stronger rainfall due to the upper vorticity, water vapor divergence in the whole layer and atmospheric precipitable water, which is basically consistent with the results of Zhang Qiang et al. [47]. Different from the previous focus on the characteristics and formation mechanism of rainfall, this paper also investigates the characteristics of floods caused by rainfall in different weather types. The research results provide a scientific basis for the prediction and warning of rainstorm floods and the quantitative assessment of their impact.

Many scholars have pointed out that in addition to the large-scale circulation, the mesoscale systems are the direct producers of heavy rainfall [48,49], so there is still a lot of work to be done on the causes of flood-causing rainfall, such as the formation and evolution mechanism of mesoscale weather systems. How does the mesoscale system interact with the large-scale circulation to affect the flood-causing rainfall in the SAYR? In addition, the topography of the SAYR is complex. This study is based on the daily precipitation data of 12 national meteorological stations in the SAYR, which cannot fully reflect the local microclimate characteristics. With the continuous extension of regional encrypted meteorological data, it is of great importance to reveal the synoptic mechanism of the refined rainfall process at the spatio-temporal scale for accurate flood disaster prevention and decision making.

## 5. Conclusions

In this study, we have undertaken a synoptic classification of the flood-causing rainfall process in the SAYR and studied the corresponding flood characteristics. The main findings are as follows:

1. The flood-causing rainfall in the SAYR can be classified into westerly trough type (type I), low vortex shear type (type II) and subtropical high southwest flow type (type III). In type I, the circulation is relatively flat and fluctuating at the mid- and high-latitudes,

and the Indo-Myanmar depression is relatively strong, causing the warm and wet flow to be transported into the SAYR, where it converges with cold air to form continuous rainfall. In type II, the high-pressure ridge near the Ural Mountains and the Sea of Okhotsk and the low trough over Lake Balkhash, which is between the two highs, are stronger than in other types in the same period of the year. The low vortex is generated at the lower levels and is accompanied by the shear line. The shear line moves eastwards to cause rainfall in the SAYR. In type III, the Balkhash Lake trough is strong, and the northerly wind behind the trough leads the cold air to converge with the southwesterly warm and wet flow around the WPSH periphery, producing heavy rainfall. In the three types, the SAH is stronger and extends more eastward than the other types in the same period of the year, the upper-level jet expands northward, the WPSH in the type II and type III is stronger and extends more westward than the other types in the same period of the year, but that is eastward in type I, and the effect is not obvious.

2. After statistical analysis of the physical quantities, it is found that the dynamic, water vapor, and thermal conditions of the three weather types are favorable. Heavy rainfall occurs in areas with low-level convergence, high-level negative vorticity, low-level positive vorticity, convergent water vapor flux in the whole layer, a certain atmospheric precipitable water, and low-level cold advection. The water vapor condition and dynamic condition of the large flood are better than those of the medium and small floods, but the thermal condition is insufficient, which only shows certain indicative significance at specific levels.

3. From the perspective of flow increase and process flood volume, type I has a long duration and relatively small flood volume, and the flood peak and cumulative flood volume corresponding to small flood are the smallest among all weather types. Type II has a short duration and large flood volume. Type III also shows the characteristics of a short duration, with the mean flood peak and cumulative flood volume of the large flood being the largest of all weather types.

**Author Contributions:** Conceptualization, B.Y. and L.J.; methodology, validation, B.Y.; investigation, writing—original draft preparation, L.J. and J.L. (Jing Liu); writing—review and editing, J.L. (Jifeng Liu) and C.Y. All authors have read and agreed to the published version of the manuscript.

**Funding:** This research was funded by the National Key Research and Development Program of China (Grant 2021YFC3201104-02), the Yellow River Water Scientific Research Project of the National Natural Science (Grant U2243229), and the Science and Technology Research Foundation Program of Henan Province (Grant 232103810095).

**Data Availability Statement:** The data presented in this study are available on request from the corresponding author.

**Acknowledgments:** The authors acknowledge the National Meteorological Information Center (NMIC) of the China Meteorological Administration for providing the daily observational rainfall data (http://data.cma.gov.cn/ (1 March 2023)), the National Oceanic and Atmospheric Administration (NOAA) for providing NCEP-1 data (https://psl.noaa.gov/data/gridded/data.ncep.reanalysis.html (1 March 2023)). The authers are grateful to the editors and anonymous reviewers for their insightful comments and suggestions.

**Conflicts of Interest:** The authors declare no conflict of interest.

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
