# Peer review of "Synoptic Analysis of Flood-Causing Rainfall and Flood Characteristics in the Source Area of the Yellow River"

_water, doi:10.3390/w16060857_

Round 1

Reviewer 1 Report

Comments and Suggestions for Authors

Proper description of the meteorological and hydrological analysis based on the identification of the sources for the weather variability for the intended use (Yellow River).

The paper intended to establish relations between the rainfall and floods by historical data of rainfall (1961 – 2000) – obtained from meteorological stations (12) and an hydrological station – and of floods (60 years), aiming to establish correlation between different level of floods with rainfall events that are considered to be the origin for those flood events.

In the case of historical data from several decades, knowing the natural evolution of measurement means and technologies, it is considered convenient that the authors can provide some more specific information about the nature and relationship of the data series, the instrumentation that gives them gave rise to, its traceability, and other elements that ensure the quality of the data used to ensure that the information is based on robust data. It should be noted that there is a complete lack of information regarding the data that supports the study.

The approach developed, which instead of following the traditional approach that focus on short-term rainfall analysis develops an approach that combine rainfall (long term) and flood to study correlation of the phenomena, considering also the impact of other influence quantities, can be considered as innovative.

It appears that the vast majority of authors cited in the bibliography are of a single geographic origin, even given the fact that the article is dedicated to a study aimed at a given region and applicable to the Yellow River, it could be useful to have, in some aspects of the discussion, perspectives from authors from other origins.

Reviewer 2 Report

Comments and Suggestions for Authors

This manuscript proposes three meteorological conceptual models of the  meteorological conditions conducents to floods conditions in the source waters of the Yelow River. These are based in gauge data from an hydrological station, precipitation data fom meteorological stations in the region and  NCEP-NCAR data from 1961 to 2020. The models, westerly trough type, low vortex shear type and subtropical high southwest flow type were identified using weather type methods. The mechanisms involved in causing flood in each of the conceptual models were investigated in terms of loand connected to three different flood levels in terms of  low level and high level convergence, vorticity, water vapor convergence,precipitable water and temperature advection. The westerly trough type has long duration and small flood volume and the others short duration and large flood volume.

General Comments.

1- Some aspects of the methodology and the results obtained are interesting, however, the methodology is not well described and some details are missing. 2- Many sentences in the paper have not an univocal meaning. 

Specific Comments. 

1- In the very Abstract, the sentence 'The Ural Mountains and the Okhotsk Sea high are stronger than the same period of the year in the low vortex shear type' is hard to understand

2-  In the Data and Methods, Definition of Rainfall Proceess, the paragraph starting  'method of judging the rainfall process..as the start date of the flood causing rainfall process' is very obscure and needs rewritting.

3- In the Method section it is said: 'The flood causing rainfall was classified according to weather types. Was this classification a subjective or objective one?. How many events were missclassified? Some quantification is needed.

T4- In the Method section it is written '    In addition, the violin plot method was used to analyze the dynamics, water vapor and instability conditions of each rainfall process' without any explanation of how a violin plot is drawn and how it was originnally proposed. But the violin plot is not so common, for instance is not explained in many books on 'statistical methods for atmospheric sciences' therefore some brief description of the basis of this method  are needed.

5- There are many sentences in the Results that have been written following the pattern of the two previously mentioned in the Abstract and in the Methods, that need a new paraphrasing.

6- Why is the section  4 called  Discussing  and not Discussion, as is the common use?. 

7- There are some misused capital letters as for instance 'In General..'

8-I find the Discussion section fine and the Conclusions interesting, but there is a lacking of details in the Method section, and a lack of clarity in the drafting of some sentences of the Results section that make the paper not ready for inmediate publication in Water.

Comments on the Quality of English Language

There is a lack of clarity in the drafting of some sentences of the Abstract, the Data and Methods, and the Results section that need to be fixed.

Round 2

Reviewer 2 Report

Comments and Suggestions for Authors

The paper has improved considerably, but there are still a couple of questions on the english expression pending. See please comments on the english language below.

Comments on the Quality of English Language

The paper has improved considerably, due to the corrections introduced by the 

authors. Only two question previously pointed by the authors are not well 

addressed. 

1- The sentence in the abstract ' 

the Ural

Mountains and the Okhotsk Sea are stronger compared to the same period, 

which is the same period?

2- The sentence in the deffinition of rainfall processes nedd still some editing

'Secondly ... at most one day of ineffective rainfall.'  The meaning of this sentence is still unclear and its english expression misleading.

I think the authors must pay attention to these two details, to avoid academic criticisms from their colleagues in the future,
